# Is the Absence of Manual Lymphatic Drainage-Based Treatment in Lymphedema after Breast Cancer Harmful? A Randomized Crossover Study

**DOI:** 10.3390/jcm13020402

**Published:** 2024-01-11

**Authors:** Iria Da Cuña-Carrera, Mercedes Soto-González, Rocío Abalo-Núñez, Eva M. Lantarón-Caeiro

**Affiliations:** 1Clinic Physiotherapy Group, Galicia South Health Research Institute, Servicio Galego de Saúde, 36312 Vigo, Spain; iriadc@uvigo.es (I.D.C.-C.); rocioabalo@uvigo.es (R.A.-N.); evalantaron@uvigo.es (E.M.L.-C.); 2Faculty of Physiotherapy, Campus A Xunqueira s/n, University of Vigo, 36005 Pontevedra, Spain

**Keywords:** manual lymphatic drainage, breast cancer, lymphedema

## Abstract

(1) Background: Manual lymphatic drainage (MLD), included within the complex decongestive therapy, as a therapy for the treatment of lymphedema has raised controversy about its benefits for lymphedema after breast cancer. The aim of this research is to test the effects of MLD on lymphedema after breast cancer during the treatment maintenance phase. (2) Methods: A randomized, single-blinded, controlled crossover trial was conducted to analyze the effects of a manual lymphatic drainage intervention compared to a control group without MLD intervention for the treatment of lymphedema. Arm volume measured by circumference measurement, subcutaneous tissue thickness measured by ultrasound, and the sensation of pain, heaviness, and swelling were evaluated as outcome measures. (3) Results: For the control group, an increase in volume was found in some of the circumference and subcutaneous tissue thickness measurements, in addition to a worsening of arm pain, swelling and heaviness. (4) Conclusion: The absence of treatment based on MLD in lymphedema after breast cancer worsens volume measurements, as well as arm heaviness. Therefore, it would be advisable to carry out this type of therapy as part of the maintenance treatment for lymphedema in breast cancer.

## 1. Introduction

Lymphedema is one of the potential complications that may occur after breast cancer. Risk factors for its occurrence are mastectomy, axillary adenectomy, radiotherapy, high body mass index and the occurrence of postoperative infections [1,2,3]. Lymphedema is characterized by persistent tissue inflammation due to an abnormal accumulation of lymph in the tissues, and it affects around 15% to 30% of patients. There are studies that suggest that more than one in five who survive breast cancer will develop arm lymphedema. This condition has a detrimental impact on the survivors’ health-related quality of life, so effective strategies for improving the quality of life of breast cancer patients are needed [4,5,6].

Conservative therapies have been shown to be effective in the treatment of lymphedema, since they reduce swelling, decrease the risk of infection and slow the progression of lymphedema. The lack of treatment could lead to an accumulation of adipocytes and fibrocytes in the affected areas, thus generating complications such as skin infections, reduced immunity or decreased functionality [7].

Complex decongestive therapy is a set of techniques that seek to treat lymphedema conservatively as described in the recent international consensus of the International Society of Lymphology published in 2020 [8]. The first phase of complex decongestive therapy aims to reduce cutaneous edema and the second phase attempts to preserve and optimize the results obtained [7,8,9,10].

The first phase consists of skin care, manual lymphatic drainage, muscle pumping exercises and compressive techniques typically applied with multilayer bandages. The second phase consists of low-stretch compression, skin care, exercises and manual lymphatic drainage, which must be repeated as needed.

The frequency and intensity of the components of complex decongestive therapy in phase I and phase II should depend on the clinical findings on edema and the stage of lymphedema, adapting to clinical changes. It should be emphasized that phase II (or ‘stabilization phase’) represents long-term therapy [8].

In recent years, there has been debate on the efficacy of manual lymphatic drainage. So much so that it is no longer prescribed for these patients and has been replaced by recommending self-management [11,12]. In addition, it has been suggested that complex decongestive therapy is time-consuming, expensive and difficult to tolerate, and does not improve lymphatic function [13].

In contrast, in 2018, researchers such as Müller et al. [14] stated that it is a well-tolerated and safe treatment technique and it shows benefits in edema reduction. Other studies have also shown that manual lymphatic drainage is effective both at a preventive level [15], and as a postoperative rehabilitation treatment, having optimal results when combined with the other elements of complex decongestive therapy [16,17,18].

Two recently published systematic reviews [19,20] highlight the need for further experimental studies on the effectiveness of manual lymphatic drainage on lymphedema. Responding to this demand, the aim of this study is to test the effects of manual lymphatic drainage on lymphedema after breast cancer surgery, in the treatment maintenance phase.

## 2. Materials and Methods

### 2.1. Design

This was a randomized, single-blinded, controlled crossover trial, conducted to analyze the effects of a manual lymphatic drainage intervention compared to a control group without manual lymphatic drainage intervention for the treatment of lymphedema in the maintenance phase. Both groups followed the same hygiene and arm care recommendations. For this investigation, the CONSORT guideline was followed [21]. The study was conducted from March 2021 to April 2022. The study protocol was in accordance with the Declaration of Helsinki and was approved by the Institutional Ethics Committee (205-2021-3) and registered on ClinicalTrials.gov (NCT05037708). All participants were informed about the aim of the study and provided with a written informed consent form prior to participation, including a specific consent for the publication of all images, clinical data and other data included in the main manuscript.

### 2.2. Participants

Thirty-four women suffering from lymphedema after breast cancer were initially recruited and 29 of them met the inclusion criteria. The participants were in the maintenance phase of treatment after an intensive phase of treatment in which she was a reduction of volume of lymphedema. According to the guidelines of complex decongestive therapy, after two weeks of stabilization in reducing the volume of the edema, the women moved on to the maintenance phase of treatment. Inclusion criteria were (I) women included in the lymphedema treatment maintenance program through the Galician Lymphedema Association; and (II) women with secondary unilateral lymphedema (stage 2 according to the International Society of Lymphology [8]) after breast cancer surgery. Exclusion criteria were (I) women undergoing chemotherapy or radiotherapy treatment; and (II) severe systemic or neurological disease. One person had to be withdrawn from the study because of COVID complications, so 28 participants completed the study.

Sample size was calculated with the software G*Power (version 3.1.1). The study of Tambour et al. [22] was selected. The variable selected was “arm circumference” and the affected vs. unaffected arm were compared (179.2 ± 4.93 vs. 154.4 ± 2.46; effect size = 0.63). Considering a power of 0.9 and an alpha error of 0.05, an estimated sample size of 23 subjects was calculated as a minimum to be able to report consistent results. A potential 30% loss to follow-up had to be taken into account; therefore, a total of 29 participants were deemed necessary to ensure adequate power for the analysis. All participants were equally randomized to group 1 and group 2 by a person who was not involved in either the assessments or the intervention. Simple randomization was performed by using tables of random numbers with the allocation reason 1:1. Hidden allocation was used.

### 2.3. Intervention

The experimental group received treatment for 4 weeks. This treatment consisted of 1 session of Manual Lymphatic Drainage (MLD) per week. Each session lasted for 60 min, starting with shoulder and trunk drainage and arm drainage (hand included) afterwards. The MLD was applied by a physiotherapist specialized in lymphatic pathology with a standardized treatment protocol based on the Leduc Method.

The control group did not receive treatment for 4 weeks. Participants in both groups used the compression sleeve daily. This is a crossover clinical trial; therefore, women participated in both the intervention and control group, having a sufficient rest period of 2 months [23,24]. The measurements were or were not carried out the week after receiving the treatment, depending on the group.

Prior to the study, all women received 2 sessions of MLD per month through the Galician Lymphedema Association program. During the study, this program was stopped in order not to influence the research results.

### 2.4. Variable Outcomes

The measurement procedure for the analyzed variables is shown below. All measurements were collected by a blinded physical therapist with specialized training.

−Arm volume via circumferential measurements: The diameter of the upper limb with lymphedema was measured with a thin and flexible plastic tape measure at 4 anatomical points: the metacarpophalangeal joint (MCP), wrist (Wrist), 10 cm below the lateral epicondyles (BLE), and 10 cm above the lateral epicondyles (ALE) [25,26]. The patient was lying down in supine position, arm included. For this variable, three measurements were conducted at each point and the mean value was calculated.−Subcutaneous tissue thickness by ultrasound imaging: Ultrasound images of the subcutaneous tissues were collected using an ultrasound scanner (GE Logic-e 4–12 MHZ, 39 mm lineal transducer; B mode). The measurements were carried out by a physical therapist with knowledge of musculoskeletal ultrasound imaging and who was trained in measuring subcutaneous tissues. The probe was placed perpendicular to the ventral axis of the upper limb [27] and sufficient ultrasound gel was applied in order to obtain a correct image without pressing on the tissues. The subcutaneous tissue thickness was obtained from the distance from the upper limit of the skin/subcutaneous tissue to the lower limit of the muscle fascia [28]. These measurements were taken in 2 locations: 10 cm above the elbow lateral epicondyles (ALE) and 10 cm below the elbow lateral condyles (BLE) [29]. The patient was lying down in supine position, arm included. Three measurements were performed at each point and the mean value was calculated. The women’s skin was marked so that the measurements were always taken in the same place, using the on-screen Calipper provided by the ultrasonography equipment.−Assessment of heaviness, pain, and tension on the upper limb: For these variables, a visual analogue scale (VAS) was used from 0 to 10 points, with 0 being non-existent heaviness, pain or tension and 10 being maximum heaviness, pain or tension [22,30,31]. Women indicated on a 100 mm line numbered from 1 to 10 how they felt their heaviness, pain and tension within the last three days before the measurement was taken [32].

### 2.5. Statistical Analysis

All statistical analyses were carried out using the Software SPSS for Macintosh (version 25.0, Chicago, IL, USA). Demographic data were analyzed using a two-sample independent t-test to detect between-group differences. The comparison between pre and post-test measurements (circumferential measurements, subcutaneous thickness, and assessment of heaviness, pain and tension) and between groups was calculated using repeated-measure analysis of variance, with an intra subject factor (experimental group versus control group). The effect size (ES) was calculated for all variables using Cohen’s d and classified as trivial (d < 0.2), small (0.2 ≤ d < 0.5), medium (0.5 ≤ d < 0.8) or large (d ≥ 0.8). The significance level was set at *p* < 0.05 [33].

## 3. Results

Twenty-eight eligible women met the criteria and agreed to participate in the study. Fourteen women were randomized to each group. The dropout rates and specific reasons for dropouts are presented in Figure 1.

The sample characteristics are displayed in Table 1. No significant demographic or clinical differences were observed between the two groups.

Concerning variables, results of the comparisons between pre- and post-test measurements and between the control and the experimental group are detailed below. Regarding the circumferential measurements, results are shown in Table 2.

As detailed in Table 2, measurements MCP, Wrist and BLE did not show any significant differences neither between groups nor between pre and post-test. However, for measurement ALE, subjects in the control group showed a significant increase in the perimeter in the post-test in comparison with the pre-test (*p* < 0.001; ES = 0.17, small) and in comparison with the experimental group (*p* = 0.017; ES = 0.08, small), which did not show any differences between pre and post-test (*p* > 0.05).

As for the ultrasonography, results of the two measurements for subcutaneous tissue thickness are described in Table 3.

As shown in Table 3, for measurement ALE, thickness decreased post intervention significantly in comparison with the pre-intervention measurement in both the control (*p* = 0.031; ES = 0.13, small) and experimental group (*p* < 0.001; ES = 0.22, medium), whereas there were no differences between groups. Nevertheless, for measurement BLE, the control group demonstrated a significant increase in thickness in the post-test measurement in comparison with the experimental group (*p* = 0.046; ES = 0.26, small), but no differences were found between the pre-measurement and post-measurement in the control group.

Regarding variables measured with the VAS scale (heaviness, tension and pain on the upper limb), results of the comparisons are shown in Table 4 through the percentage of change between pre and post.

As for heaviness measured with the VAS scale, no differences were reported between pre- and post-test measurements. However, significant differences were found between the control and experimental group (*p* = 0.001; ES = 0.49, medium), supported by the percentage of change (49.08% vs. −21.56%). This showed that women in the control group experienced increased heaviness post intervention but the experimental group experienced decreased heaviness in that moment.

On the other hand, no significant differences were found in tension or pain, neither between groups nor between pre and post treatment. Considering the percentage of change in these variables (no significant differences), the tension seemed to increase in the control group but decreased slightly in the experimental group (32.18% vs. −0.88%) and the pain increased in both the control and experimental group, but more in the control group (44.04% vs. 18.37%).

## 4. Discussion

The aim of this study was to determine the effects of a physical therapy treatment based on manual lymphatic drainage in women with lymphedema after breast cancer in the lymphedema maintenance phase of the treatment.

As breast cancer overall survival continues to improve, quality of life measures become increasingly important among long-term survivors. Breast cancer-related lymphedema is a common treatment-related morbidity, and several risk factors have been identified to be related to axillary surgery and radiation techniques [34]. As indicated above, there is some discrepancy about the effectiveness of this technique for the prevention [35,36] and treatment [16,27,28,29] of lymphedema, despite being included among the therapies recommended by the most recent International European Consensus on Lymphedema in 2020, within the complex decongestive therapy [8]. Also, the patients that do not improve with conservative treatment should be considered to undergo a surgical treatment of lymphedema, and is these case preoperative and postoperative complex decongestive therapy is mandatory to improve the outcomes [8,37,38]

We focused on the maintenance phase or phase 2 of the treatment for lymphedema, with the aim to preserve and optimize the results obtained in the intensive phase or phase 1 [8]. In our study, we found that the control group, in which the subjects stopped receiving the manual lymphatic drainage sessions, experienced an increase in the circumference and subcutaneous tissue thickness at some measurement points. However, at other measurement points we found no difference in the changes that occurred between the intervention and control group. This could be due to the short intervention/non-intervention time in this study (only 1 month), which is perhaps not long enough to observe the effects of not performing manual lymphatic drainage in the edema maintenance phase. In addition, the washout period in this crossover clinical trial was only two months.

In relation to the volume measurement, which is related to these variables, much of the literature indicates that manual lymphatic drainage does not improve this parameter, as stated in the review by Yan Lin et al. [39] and in the trials by Tambor et al. [22], Sen et al. [31] and De Vrieze et al. [40]. It should be pointed out that, in the studies mentioned above [22,31,40], manual lymphatic drainage was not carried out alone, but it was combined with other therapies such as therapeutic exercise. Therefore, therapeutic exercise could have a positive effect on its own, as Kilbreath et al. [41] concluded, and lymphatic drainage does not have a greater effect. Women with lymphedema have shown a problem with adherence in these exercise therapy programs [42,43]. For this reason, we analyzed the effects of MLD without any comparison to study the effects of this method. Women did not carry out any physical exercise program in any of the groups, and this could explain that, in some measurement points related to perimeter and ultrasound, a worsening of the lymphedema is observed if manual lymphatic drainage sessions are discontinued. This coincides with Zimmermann et al. [18], who pointed out that the performance of manual lymphatic drainage, in comparison with the absence of treatment, had a protective effect against the appearance of lymphedema after breast cancer.

At this point, it is important to remark that the best way to examine the volume and structural changes of lymphedema is not known to date. Volumetry uses water displacement as a form of measurement. However, it cannot differentiate the volume of the subcutaneous tissue where edema would be found from deep structures, such as muscles and bones [44,45]. The same occurs with the arm circumference measurement, in addition to the fact that errors can be made due to tape pressure, marked measuring points or an inadequate angle in relation to the longitudinal axis of the limb [43,46].

In this research, we used circumference measurement as a measurement for the change in edema volume, even though it has disadvantages, but we also added a newer evaluation technique (ultrasonography). Ultrasonography does allow us to obtain the measurement of the subcutaneous tissue, differentiating it from other deeper structures. Nevertheless, this method is limited, since only the thickness of a localized point can be obtained, without evaluating the entire limb circumference [47]. The evaluation of subcutaneous tissue thickness can be a useful tool as a diagnostic measurement method for lymphedema [48] and it correlates with other measurement methods such as volumetry and circumference measurement [25].

In the ultrasonography measurement, it can be observed that those patients who have not received manual lymphatic drainage have experienced an increase in subcutaneous tissue thickness in the forearm area (ultrasound measurement ALE), while thickness is practically the same in those patients who have received it. Precisely, this indicates that manual lymphatic drainage contributes to the maintenance of lymphedema, which is the objective pursued in this maintenance phase. Therefore, leaving these women without this treatment could mean an aggravation of lymphedema, as shown in our results.

At this same measurement point, the circumference measurement (measurement BLE) detected an increase in lymphedema, although not significant, which could indicate that ultrasound is more accurate than tape measurements when evaluating changes in lymphedema [48].

Other variables studied for the evaluation of the effect of manual lymphatic drainage on lymphedema after breast cancer were pain, heaviness and swelling by using the visual analog scale. Significant changes were found in heaviness, which increased in the control group and decreased in the experimental group. In addition, although there were no significant differences, pain and tension showed an increase in the percentage change pre and post treatment in the control group. Therefore, our findings indicate that the performance of manual lymphatic drainage has a positive effect on these variables. The recent review by Lit et al. [38] also found a positive effect of manual lymphatic drainage on pain.

Given these results, we could hypothesize that, despite having no clear effect on the volume of lymphedema, manual lymphatic drainage could make histological changes in the tissue that improve variables such as arm heaviness, pain or tension. It is believed that manual lymphatic drainage increases lymphatic drainage through the stimulation of superficial lymphatic contraction and the diversion of lymphatic fluid to adjacent functioning lymphatic systems [11,49].

Manual lymphatic drainage influences the absorption of macromolecules that make up edema [50]. Therefore, it would have a limited effect on reducing edema volume compared to other techniques. Thus, it does not cause hemodynamic alterations even in patients with cardiac pathologies [51]. In relation to the effect of manual lymphatic drainage on the lymphatic pathways, research has been carried out through lymphography with indocyanine green, and it has been found that the call maneuvers proposed by the Leduc Method (drainage method not included in the report analysis) increase the frequency of observation of collateral lymphatic areas and pathways in women undergoing surgery for breast cancer. Lymphoscintigraphy is an accepted imaging analysis for the diagnosis of lymphatic transport dysfunctions by quantifying the transit time of radionuclide transport from an injection site or its accumulation in draining lymph node [52].

Another fundamental aspect that must be taken into account in the evaluated literature is the way in which manual lymphatic drainage is performed. There are different schools of lymphatic drainage (Vodder, Casley-Smith, Leduc, and Foldi) [53] and there may be differences in drainage results. However, all these different methods coincide in fundamental aspects, such as that manual lymphatic drainage should be performed by a trained physiotherapist and requires a long application time (from 40 min to 1 h) [53,54]; also, slow repetitive hand movements are performed, gently massaging along the anatomical lymphatic pathways over the affected areas that try to stimulate lymphatic flow [11,53]. The reviews checked [19,35,39,49] do not specify the type of drainage performed or do not differentiate between the results depending on the type of drainage. We believe that this variable should be taken into account for future studies.

The limitations of this study include the fact that no differentiation was made between mild, moderate, or severe edema. In addition, as this was a maintenance phase, the time elapsed between surgery and cancer was variable across women. We also believe in the need to test the comparison of manual lymphatic drainage treatment and an exercise therapy, in order to know whether the benefits are similar and if manual lymphatic drainage could be as valid an option as therapeutic exercise, since adherence to therapeutic exercise is not always easy and, therefore, not always carried out.

## 5. Conclusions

The absence of treatment based on manual lymphatic drainage in lymphedema after breast cancer causes worsened volume measurements, as well as arm heaviness. Therefore, it is advisable to perform it during the maintenance phase.

However, in our study we only evaluated the effect of therapy based on manual lymphatic drainage without combining it with other techniques that make up complex decongestive therapy, such as therapeutic exercise, which could increase the magnitude of the effects obtained with each of the techniques in isolation.

## Figures and Tables

**Figure 1 jcm-13-00402-f001:**
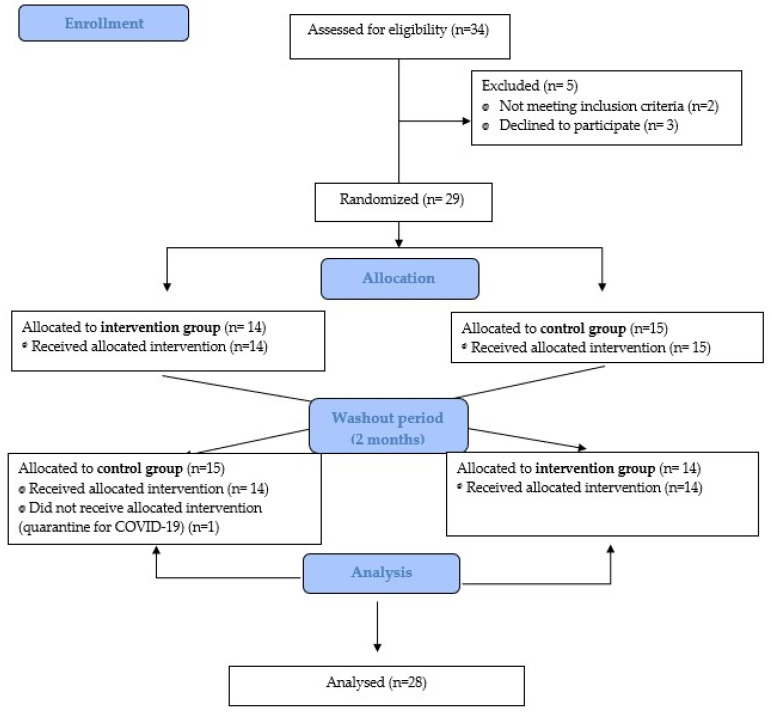
Flow diagram.

**Table 1 jcm-13-00402-t001:** Demographic characteristics.

Sample Characteristics	Total *n* = 28	Groups					
		Group 1	Group 2	Mean Difference (SD)	*p*-Value	95% CI for Difference	
		*n* = 14	*n* = 14			Lower	Upper
Age, mean (SD), years	59.89 (10.19)	59.57 (10.861)	60.21 (9.870)	6.43 (3.92)	0.87	−8.71	7.42
BMI, mean (SD), Kg/m^2^	26.64 (4.76)	26.98 (4.59)	26.29 (5.08)	0.69 (1.83)	0.71	−3.07	4.45
Weight, mean (SD), Kg	69.2 (14.45)	72.42 (15.24)	66.09 (13.4)	6.34 (5.43)	0.25	−4.81	17.48
Height, mean (SD), cm	160.96 (7.36)	163.43 (7.18)	158.50 (6.93)	4.93 (2.67)	−0.076	−0.55	10.41

BMI, Body Max Index; CI, Confident interval; SD, Standard Deviation.

**Table 2 jcm-13-00402-t002:** Results of the comparisons pre–post measurements and groups in the circumferential measurements.

	MCP (Mean ± SD)		Wrist (Mean ± SD)		BLE (Mean ± SD)		ALE (Mean ± SD)	
	Pre	Post	Pre	Post	Pre	Post	Pre	Post
CG (*n* = 28)	19.35 ± 1.22	19.48 ± 1.24	17.34 ± 1.9	17.53 ± 2.09	26.49 ± 4.17	26.7 ± 4.17	31.46 ± 4.17	32.19 ± 4.38 *^,#^
EG (*n* = 28)	19.42 ± 1.31	19.49 ± 1.53	17.6 ± 2.19	17.49 ± 2.04	26.7 ± 4.17	26.52 ± 4.67	31.69 ± 4.34	31.85 ± 3.95

ALE: 10 cm above the lateral epicondyles; BLE 10 cm below the lateral epicondyles; CG: control group; EG: experimental group; MCP: metacarpophalangeal joint; SD: standard deviation. * Significant differences between pre and post measurements. ^#^ Significant differences between CG and EG.

**Table 3 jcm-13-00402-t003:** Results of the comparisons pre–post measurements and groups in the ultrasonography measurements for subcutaneous tissues.

	ALE (Mean ± SD)		BLE (Mean ± SD)	
	Pre	Post	Pre	Post
CG (*n* = 28)	0.57 ± 0.23	0.54 ± 0.22 *	0.77 ± 0.29	0.87 ± 0.33 ^#^
EG (*n* = 28)	0.58 ± 0.21	0.53 ± 0.24 *	0.79 ± 0.28	0.79 ± 0.27

ALE: 10 cm above the lateral epicondyles; BLE: 10 cm below the lateral epicondyles; CG: control group; EG: experimental group; SD: standard deviation. * Significant differences between pre and post measurements. ^#^ Significant differences between CG and EG.

**Table 4 jcm-13-00402-t004:** Results of the comparisons between pre and post measurements and groups with regard to heaviness, tension and pain.

	Heaviness (Mean ± SD)			Tension (Mean ± SD)			Pain (Mean ± SD)		
	Pre	Post	Δ	Pre	Post	Δ	Pre	Post	Δ
CG (n = 28)	3.28 ± 2.51	4.89 ± 2.62	49.08%	2.89 ± 2.93	3.82 ± 2.96	32.18%	1.93 ± 2.27	2.78 ± 3.09	44.04%
EG (n = 28)	4.5 ± 2.34	3.53 ± 2.95 ^#^	−21.56%	3.42 ± 3.06	3.39 ± 2.68	−0.88%	1.96 ± 2.23	2.32 ± 2.95	18.37%

CG: control group; EG: experimental group; SD: standard deviation. Δ Percentage of change (%) between pre and post measurements. ^#^ Significant differences between CG and EG.

## Data Availability

No new data were created or analyzed in this study. Data sharing is not applicable to this article.

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
