# Peer review of "Is the Absence of Manual Lymphatic Drainage-Based Treatment in Lymphedema after Breast Cancer Harmful? A Randomized Crossover Study"

_jcm, 2024, doi:10.3390/jcm13020402_

Round 1

Reviewer 1 Report

Comments and Suggestions for Authors

JCM-2770309-peer-review-v1:  Is the absence of manual lymphatic drainage-based treatment in lymphedema after breast cancer harmful” A randomized crossover study.

Recognizing the potentially devastating effects of post-treatment lymphedema in breast cancer patients, the authors have sought to contribute to the literature addressing whether manual lymphatic drainage-based (MLD) treatment provides additional benefits for patients undergoing the second phase of complex decongestive therapy.  The efficacy of this intervention was tested using a randomized crossover trial design, with several morphometric and sensory variables measured. Authors report positive effects of four weeks of weekly MLD treatment.

This is a straight-forward design and analysis of an important potential treatment option for patients with lymphedema.  But there are several issues that need to be addressed.

1.     A main question is related to issues raised in both the Introduction and the Discussion.  In the introduction the authors acknowledge that the MLD has largely been replaced by self-massage.  It was not stated whether study participants engaged in self massage, although the design suggests they did not.  Why was there no direct comparison (additional groups) evaluating the MLD vs. self-massage vs no massage-related treatment?  This comparison would have addressed the current approach of eliminating the more time-consuming and costly MLD and substituting a potentially equally effective, but less resource-dependent alternative.

Similarly, in the Discussion, the authors acknowledge that physical exercise programs might have sufficient therapeutic efficacy and that MLD might not add to the efficacy of post-treatment lymphedema interventions.  Why was this observation not addressed in the design of the current study?  It is difficult to accept the final fairly universal conclusion of the report, as offered in the Abstract and Discussion, given the lack of these direct comparisons.  The authors acknowledge the need for the broader comparisons.  Without them, this paper is somewhat compromised.

2.      More specific points:

a.     In several places the manuscript needs editing for clarity, decreased redundancy, etc.

b.     Sample size:  The woman who developed COVID was dropped from the final study, as made clear in the results presented in the various tables.  Please change the number from 29 to 28 in the description of the study, perhaps acknowledging in the participants description that one person had to be dropped because of COVID complications.

c.     Volume measurements: Instead of numbering the volume measurements # 1-4, it would be clearer throughout the paper to use appropriate abbreviations that are relevant to the actual measurement.  For example, MCP as an abbreviation for metacarpophalangeal joint is much more informative than ‘measurement 1’.

d.      Table 4 and the prose in the article are confusing.  Although there is a symbol (*) for significant pre-post treatment measurement differences, this symbol does not appear anywhere in the Table.  And although the differences in tension and pain were not statistically different, the authors go on to discuss them – both in the Results section and the Discussion – as if they were significant.

Comments on the Quality of English Language

In general, the English is fine in this paper.  There are, however, several grammatical errors and odd phrases.  A thorough and careful edit is recommended.

Author Response

First, we want to thank you for your time and knowledge for reviewing the paper. We sincerely believe all the corrections and suggestions have been carried out and hope the manuscript will increase in quality and interest. 

Please find below all the answers for every comment. All changes were marked in red color in the paper. 

A main question is related to issues raised in both the Introduction and the Discussion.  In the introduction the authors acknowledge that MLD has largely been replaced by self-massage.  It was not stated whether study participants engaged in self-massage, although the design suggests they did not.  Why was there no direct comparison (additional groups) evaluating the MLD vs. self-massage vs no massage-related treatment?  This comparison would have addressed the current approach of eliminating the more time-consuming and costly MLD and substituting a potentially equally effective, but less resource-dependent alternative.

Thakns for the comment. The sample is this study was not instructed in the self-massage. However, women came regularly to MLD sessions performed by a physiotherapist. In this research we focused on the effects of MLD versus no treatment, but we consider the purpose of analyzing self-massage as very interesting for future research. Additionally, previous research has found difficulties in complying with home treatments, which could lead to therapeutic failure.

Similarly, in the Discussion, the authors acknowledge that physical exercise programs might have sufficient therapeutic efficacy and that MLD might not add to the efficacy of post-treatment lymphedema interventions.  Why was this observation not addressed in the design of the current study?  It is difficult to accept the final universal conclusion of the report, as offered in the Abstract and Discussion, given the lack of these direct comparisons.  The authors acknowledge the need for the broader comparisons.  Without them, this paper is somewhat compromised.

We completely agree with the suggestion of the reviewer and as mentioned in the discussion section exercise therapy has been reported positive effects in the treatment of lymphoedema.  However, previous studies have shown a problem of adherence in these exercise therapy programs. For this reason, we analyzed the effects of MLD without any other comparisons to study the effects of this method. Nonetheless, we fully agree exercise therapy could be a great intervention when addressing lymphoedema and future research will consider this therapy.

More specific points:

In several places the manuscript needs editing for clarity, decreased redundancy, etc.

After received the revision of the paper, the manuscript was reviewed by a professional translator, to improve the English language. We corrected and reviewed the full paper. 

Sample size: The woman who developed COVID was dropped from the final study, as made clear in the results presented in the various tables.  Please change the number from 29 to 28 in the description of the study, perhaps acknowledging in the participants description that one person had to be dropped because of COVID complications.

Thanks for the comment. We clarificated this point, and the next sentence were addended in the participant’s section:

One person had to be dropped because of COVID complications, so 28 participants completed the study.

Volume measurements: Instead of numbering the volume measurements # 1-4, it would be clearer throughout the paper to use appropriate abbreviations that are relevant to the actual measurement. For example, MCP as an abbreviation for metacarpophalangeal joint is much more informative than ‘measurement 1’.

Thanks for the comment. We agree with these appreciations, and we changed the names of the measurements.

Circumference measurements

Measurement 1: MCP ( metacarpophalangeal joint)

Measurement 2: Wrist

Measurement 3: BLE (below lateral epicondyles)

Measurement 2: ALE (Above lateral epicondyles)

Ultrasonography measurements

Measurement 1: ALE (Above lateral epicondyles)

Measurement 2: BLE (below lateral epicondyles)

Table 4 and the prose in the article are confusing. Although there is a symbol (*) for significant pre-post treatment measurement differences, this symbol does not appear anywhere in the Table.  And although the differences in tension and pain were not statistically different, the authors go on to discuss them – both in the Results section and the Discussion – as if they were significant.

We delete this sentence in the foot table because there aren’t significant pre-post treatment measurement differences. We remark in the result and discussion section that the results about tension and pain were not statistically different.

Other changes

We added the structure abstract (200 words)

We added some paragraphs in the introduction section and in the Discussion section to increase the word count of the main text as suggested per editor. 

Reviewer 2 Report

Comments and Suggestions for Authors

Intervention paragraph should be written with past tense

In table 3 measurement 2 there is no indication whether it is significant or non significant

Line 189. “but no differences were found  between the moment.” This sentence should be clarified

In table2 and table 4 no * and # are showing significant and non-significant comparisons

line 206: "On the other hand, no significant differences were found in tension or pain, neither between groups nor between the moment" here and elsewher in the text the word "moment/moments" should be better clarified for the readers

Comments on the Quality of English Language

In several senteces there is no verb

Author Response

First, we want to thank you for your time and knowledge for reviewing the paper. We sincerely believe all the corrections and suggestions have been carried out and hope the manuscript will increase in quality and interest. 

Please find below all the answers for every comment. All changes were marked in red color in the paper. 

Intervention paragraph should be written with past tense. Thanks for the correction. We have written the paragraph intervention in past.

In table 3 measurement 2 there is no indication whether it is significant or non-significant.

Line 189. “ But no differences were found between the moment.” This sentence should be clarified

We added this sentence: but no differences were found between the pre-measurement and post-measurement in the control group.” We hope this sentence will be understood better.

In table2 and table 4 no * and # are showing significant and non-significant comparisons

In the table 2 we delete * because there aren’t significant pre-post treatment measurement differences and in the table 4 this symbol didn’t exist because there were not significant different between pre-post treatment in each group.

The # symbol means significant different between control group and experimental group.

line 206: "On the other hand, no significant differences were found in tension or pain, neither between groups nor between the moment" here and elsewher in the text the word "moment/moments" should be better clarified for the readers.

Thanks for the comment. We change the sentence for better compression:

On the other hand, no significant differences were found in tension or pain, neither between groups nor between pre and post treatment.

Other changes

We added the structure abstract (200 words)

We added some paragraphs in the introduction section and in the Discussion section to increase the word count of the main text as suggested per editor. 

The manuscript was reviewed by a professional translator, to improve the English language.

Round 2

Reviewer 1 Report

Comments and Suggestions for Authors

The authors have addressed most of the issues raised in the initial review of this manuscript.

In their responses to my concerns about not addressing exercise therapy in the design and discussion, the authors presented comments about concerns about adherence for exercise therapy as an explanation for not including separate groups for comparison in the current study.  These comments should be incorporated more fully into the current discussion to clarify this point.

Author Response

Again, we want to thank you for your time and knowledge for reviewing the paper. All changes, of the second round of review,  were marked in blue color in the paper.

In discussion section, we incorporate the explanation about the poor adherence in exercise therapy programs by women with lymphoedema. This was the reason why we analyzed the effects of MLD without any other comparisons. 

Reviewer 2 Report

Comments and Suggestions for Authors

Line 29-31. The authors should clarify which type of patients are affected

The abbreviation MLD should be reported in the same manner throughout the entire manuscript (i.e. line 102 DLM)

Please revise the word “moments” throughout the entire manuscript (i.e. table 2) as recommended during 1st round of review

Line 172,173 and later in the text. What ES and small/medium stand for? This should be clarified

Line 302 “Lymphogammography” please revise with lymphoscintigraphy

In conclusions section “The absence of treatment based on manual lymphatic drainage in lymphedema after breast cancer causes worsens volume measurements, as well as arm pain, heaviness and  Tension”

however in table pain and tension were not reported as significantly modified. Please revise also acknowledging that these did not significantly improve

This study is focusing on phase II of CDT. The authors did not disclose whether during phase I all the patients reported a significant improvement. This should be clarified.

Moreover, the relevance of this issue (the evaluation of the results of CDT) as a relevant step in patients assessment should be commented in discussion section. The patients that do not improve should be considered to undergo a surgical treatment of lymphedema. Please expand discussion using the following references regarding surgical and surgical+conservative treatment of lymphedema

Executive Committee of the International Society of Lymphology. The diagnosis and treatment of peripheral lymphedema: 2020 Consensus Document of the International Society of Lymphology. Lymphology. 2020;53(1):3-19.

Ciudad P, et al. The breast cancer-related lymphedema multidisciplinary approach: Algorithm for conservative and multimodal surgical treatment. Microsurgery. 2023;43(5):427-436. doi:10.1002/micr.30990

Schaverien MV, Coroneos CJ. Surgical Treatment of Lymphedema. Plast Reconstr Surg. 2019;144(3):738-758. doi:10.1097/PRS.0000000000005993

Author Response

Again, we want to thank you for your time and knowledge for reviewing the paper. Your comments and suggestions will improve the quality of the article

Please find below all the answers for every comment. All changes of the second round of review, were marked in blue color in the paper. 

Line 29-31. The authors should clarify which type of patients are affected

Thanks for the comment. We have explained the type of patients are affected, and we have included the next sentence:

There are researches that suggest that more than one in five who survive breast cancer will develop arm lymphedema. 

The abbreviation MLD should be reported in the same manner throughout the entire manuscript (i.e. line 102 DLM)

Thanks for the correction. We have changed the abbreviation to MLD

Please revise the word “moments” throughout the entire manuscript (i.e. table 2) as recommended during 1st round of review

Thanks for the comment. We have deleted the word “moments” and we have included “pre-post measurements”

Line 172,173 and later in the text. What ES and small/medium stand for? This should be clarified

Thanks for the question. ES means the effect size witch was calculated for all variables using Cohen’s d and classified as trivial (d < 0.2), small (0.2 ≤ d < 0.5), medium (0.5 ≤ d < 0.8) or large (d ≥ 0.8) [33]. This information is in the methods section. We have included the abreviation in the methods section: Efecct size (ES) because was missed in the main document.

Line 302 “Lymphogammography” please revise with lymphoscintigraphy

Thanks for the correction. We have changed to “lymphoscintigraphy”

In conclusions section “The absence of treatment based on manual lymphatic drainage in lymphedema after breast cancer causes worsens volume measurements, as well as arm pain, heaviness and  Tension” however in table pain and tension were not reported as significantly modified. Please revise also acknowledging that these did not significantly improve

Thanks for your comment. We agree with you and we have deleted the “pain and tension” in the conclusion because we din´t find significantly change.

This study is focusing on phase II of CDT. The authors did not disclose whether during phase I all the patients reported a significant improvement. This should be clarified.

Thanks for your comment. We added this information in Methods section.  All women had improvevent in the volumen of lymphoedema in fase I of treatment.  According to the guidelines of complex decongestive therapy, after two weeks of stabilization in reducing the volume of the edema, the women moved on to the maintenance phase of treatment.

Moreover, the relevance of this issue (the evaluation of the results of CDT) as a relevant step in patients assessment should be commented in discussion section. The patients that do not improve should be considered to undergo a surgical treatment of lymphedema. Please expand discussion using the following references regarding surgical and surgical+conservative treatment of lymphedema

Executive Committee of the International Society of Lymphology. The diagnosis and treatment of peripheral lymphedema: 2020 Consensus Document of the International Society of Lymphology. Lymphology. 2020;53(1):3-19.

Ciudad P, et al. The breast cancer-related lymphedema multidisciplinary approach: Algorithm for conservative and multimodal surgical treatment. Microsurgery. 2023;43(5):427-436. doi:10.1002/micr.30990

Schaverien MV, Coroneos CJ. Surgical Treatment of Lymphedema. Plast Reconstr Surg. 2019;144(3):738-758. doi:10.1097/PRS.0000000000005993

Thanks for the suggestion. We know the importance of CDT as pre-treatment and post-treatment after a surgery. We have added the next sentence in the discussion section and we have used the suggested references.

Also, the patients that do not improve with conservative treatment should be consid-ered to undergo a surgical treatment of lymphedema, and is these case preoperative and postoperative complex decongestive therapy is mandatory to improve the out-comes [8,37,38]